# Towards Real-Time Hyperspectral Multi-Image Super-Resolution Reconstruction Applied to Histological Samples

**DOI:** 10.3390/s23041863

**Published:** 2023-02-07

**Authors:** Carlos Urbina Ortega, Eduardo Quevedo Gutiérrez, Laura Quintana, Samuel Ortega, Himar Fabelo, Lucana Santos Falcón, Gustavo Marrero Callico

**Affiliations:** 1European Space Agency, TEC-ED, 2201 AZ Noordwijk, The Netherlands; 2Research Institute for Applied Microelectronics (IUMA), University of Las Palmas de Gran Canaria (ULPGC), 35017 Las Palmas de Gran Canaria, Spain; 3Telespazio Belgium SRL, Huygensstraat 34, 2201 DK Noordwijk, The Netherlands; 4Nofima—Norwegian Institute of Food Fisheries and Aquaculture Research, NO-9291 Tromsø, Norway; 5Fundación Instituto de Investigación Sanitaria de Canarias (FIISC), 35017 Gran Canaria, Spain

**Keywords:** hyperspectral imaging, super-resolution, image processing, computational histology, remote sensing

## Abstract

Hyperspectral Imaging (HSI) is increasingly adopted in medical applications for the usefulness of understanding the spectral signature of specific organic and non-organic elements. The acquisition of such images is a complex task, and the commercial sensors that can measure such images is scarce down to the point that some of them have limited spatial resolution in the bands of interest. This work proposes an approach to enhance the spatial resolution of hyperspectral histology samples using super-resolution. As the data volume associated to HSI has always been an inconvenience for the image processing in practical terms, this work proposes a relatively low computationally intensive algorithm. Using multiple images of the same scene taken in a controlled environment (hyperspectral microscopic system) with sub-pixel shifts between them, the proposed algorithm can effectively enhance the spatial resolution of the sensor while maintaining the spectral signature of the pixels, competing in performance with other state-of-the-art super-resolution techniques, and paving the way towards its use in real-time applications.

## 1. Introduction

Image Super-Resolution (SR) reconstruction gathers a series of techniques whose purpose is to enhance the resolution of a single image or a video sequence. When a single image is the source of information for the enhancement, a number of characteristics of the image can be used to further improve the result of the process, such as features of the main object/s [1], or the used sensor [2,3], or machine learning techniques [4,5,6]. Instead, when the source is a video sequence or a series of non-identical low-resolution (LR) images of the same object, the non-redundant information between those sources can be used to enhance the process of escalation [7]. In general terms, the results of an escalation purely based on a single image are difficult to generalize. In consequence, multi-image super-resolution (MISR) is traditionally preferred for such general case, as analyzed by [8], while single-image super-resolution (SISR) can offer very good results when the number of assumptions about the image increases. The reason behind this is simple: the amount of extra information about the object is higher when multiple images are available [9].

Nevertheless, MISR has intrinsic difficulties; one of the main problems of handling multiple images is the object tracking and the selection of the most appropriate images to be combined for reconstruction. One of the approaches that has addressed both problems and is used in this work as a reference is [10], where the authors proposed a systematic approach to the selection of the best candidates for image fusion through Structural Similarity Index (SSIM) pre-comparison. This proposal also included a relatively simple but effective block matching algorithm to tackle the objects tracking along the image.

Focusing on the multi-image source case, it can be generalized that the non-redundant information contained in the LR images is most commonly introduced by sub-pixel shifts. Each individual sub-pixel shift can be the result of an uncontrolled motion of the imaging system (e.g., camera micro-vibration), an autonomous movement of the object under study, or due to a controlled motion of it, as happens on satellite remote sensing applications (satellite orbit and pointing defines the motion parameters). Taking into consideration the nature of an imaging sensor, each LR frame can be conceived as a decimated and aliased observation of the true scene [11]. Hence, MISR is possible only if sub-pixel motions between LR frames exist. This work applies to sequences of a specific imaging technique: Hyperspectral (HS) Imaging (HSI).

An HS image is a type of digital image that contains information on several spectral channels (also called bands) of the same scene, in contrast with the traditional RGB (Red-Green-Blue) image, which contains three channels, associated to red (700 nm), green (546.1 nm) and blue (435.8 nm) bands, or a monochromatic image, with only a single channel. This way, in an HS image (also called *HS cube*) each pixel corresponds to a vector containing the spectral signature of the material or substance present in the captured scene [12]. In a certain sense, this is by itself an act of gathering information about an object or scene without coming into physical contact. There are many specific cases for which HSI presents an interesting approach, such as satellite remote sensing [13], food quality assessment [14], and agriculture [15], dermatology [16]. In this publication we focus on medical applications, in particular on computational histology [17].

Despite its large advantages, the main problem of working with HSI is the high computational cost for processing HS cubes: hundreds of bands in every pixel depth imply hundreds of times larger image size. One of the main motivations of this work is to obtain a simple approach from the mathematical operation point of view. In consequence, the already critical aspect of handling HS images in terms of system memory storage, bandwidth, and computational power, will not be aggravated.

SR is particularly valuable when it is not feasible to obtain images at the desired resolution with the available sensors [18], or for solving specific deficiencies of a sensor [19]. Such is the case for HS images: the availability of very high-resolution cameras in the RGB spectra has increased dramatically in the last two decades, while the HS sensor technology has still lagged behind on this regard. One of the reasons for this is that the wavelength associated to the spectral bands imposes a limitation on the size and separation of the sensing pixels, which results in a limited density of the sensor. Another reason has been the cost of the cameras compared to the RGB mass-market ones.

The purpose of this work is to propose a method to achieve SR based on a series of different HS images of the same scene, gaining value from the combination of information from the different spectral bands for such purpose. The general problem approach will be presented, though the focus will remain on the most interesting case for computational histology applications. Several sampling factors influence the performance of the scaling algorithm, which will also be discussed in depth.

From the discussion above, we can say now that sub-pixel shift is a key concept to the MISR process. To maximize their benefit to the SR of the histological image, we will use a set-up (see Section 2.1.1) that allows us to obtain sequences of images with global sub-pixel shifts. Using global motion will simplify the motion estimation, and ultimately the execution time of the algorithm. With the optical lenses used (5×, 10×, and 20× magnification), the sub-pixel motion seen by the sensor is still deterministic and does not suffer from stochastic processes that would appear when arriving close to the diffraction limit (like in stochastic optical reconstruction microscopy [20]), but the principle of resolving them in time is the same.

The first step that is needed to be defined in an SR image reconstruction system is the observation model. Since our data source is a microscope with HSI capabilities with a motorized platform, the model expects still individual images within each sequence. Consequently, in this article, the following assumptions are considered:The input is based on continuous natural scenes, well approximated as band-limited signals.Naturally, these signals can be contaminated by the laying medium between the scene and the sensor (including the optics), or by the movement of one of the two. However, the result is, in any case, an effective motion between the sensor and the scene to be captured, leading to multiple frames of the scene connected by local and/or global shifts.There are different types of blurring effects that can affect the image in its process of going through the camera system into the image sensors. The most important one is the down-sampling of the image into pixels.These down-sampled images are further affected by the sensor noise.

In a nutshell: the frames captured by the LR imaging system are blurred, decimated, and noisy versions of the underlying true scene. Such a traditional approach is presented by [21]. In a modern system, it would be necessary to add a step afterwards that accounts for the compression applied before storage, which is used in most implementations that are limited in data storage capacity.

### Background Work

The initial work on SR was approached in the frequency domain [22], relating high- to low-resolution images through a formulation on that domain based on Continuous and Discrete Fourier Transforms, using their aliasing and shift properties. As it assumes noise-free acquisition and no blurring effects, some extensions have focused fundamentally on handling more complicated observation models, like [23].

Other lines of research focus on slicing the general image problem into single objects, enhancing the resolution of each one of them individually. Using motion information, detecting and tracking individual objects over the time, and then combining all the information obtained, has been proven to be a powerful technique, although it is limited to movements in the same 2-D plane [24]. The result is that a tracked object remains sharp while the other non-tracked objects blur out.

More and more during the last decade, the SR problem has evolved from being addressed through example-based methods, to using almost exclusively machine learning techniques [25], focusing mostly on a single image source [26]. A few works have tried to combine the advantages of single-image SR based on deep learning with the benefits of information fusion offered by multiple-image reconstruction, from which we can highlight the solution EvoIM by [27]. In that study, a genetic algorithm is employed to optimize the hyper-parameters that control the imaging model used in another study, *Fast and Robust Super-Resolution* [28], and to evolve the convolution kernels employed in its proposed processing system. They showed that the reconstruction process can be effectively adapted to different image systems and conditions.

Traditionally, when the resolution of HS images was not enough for the target application, the approach has been to individually scale separately each of the bands. This has several drawbacks, among which the following must be highlighted:Very large processing needs as the number of bands increases, which is not always affordable in embedded applications.Not using inter-band information to improve the result, while an effective SR reconstruction would improve with frequency aliasing in the LR images [29].

Recently, some researchers have tried to enhance HSI spatial resolution through the fusion of high-resolution (HR) low-spectral content images with LR high-spectral content images. A particularly interesting approach is the the one proposed by [30], who propose to reformulate the fusion problem as the estimation of a spectral dictionary from the LR HS image and the regression coefficients of both images. The results obtained involve a high computational cost, but encouraging for SISR. Another interesting approach that follows the same line is the one of [31], where the authors use the multidimensional structure of the HSI and MSI to propose a coupled tensor factorization framework that can effectively overcome the main problems that emerge in their application. The results are quantitatively compared with other state-of-the-art algorithms, and that is the reason why we have considered this publication as an important source of comparison data with the algorithm we propose.

Another interesting approach, but always focused on single-image and image-class specific machine learning, is the one from [32], who focuses on lowering the distortion of the spectral signature during the spatial resolution enhancement via constraining spatially the LR images while the deep spectral difference convolutional neural network learns the end-to-end spectral difference mapping between LR and HR HS images.

In this work, we will focus on the simplicity of the algorithm with the objective of approaching fast execution times for embedded applications. For the particular case of medical applications, we will demonstrate that the algorithm remains robust across a large portion of the studied spectrum and escalates well from twice to four times the spatial resolution per band of the original image with respect to a Bilinear Interpolation (BI) [33]. A remote sensing dataset will be used for the purpose of direct algorithmic comparison. Moreover, we will present examples of how the spectral signature is improved during image restoration, paving the way to its application in the field of histology.

## 2. Materials and Methods

In the general MISR reconstruction case there are three major steps:Capturing a sequence of images from the same scene with sub-pixel shifts between each of them (*acquisition*).Estimating the sub-pixel shift between the image taken as reference for reconstruction and the rest of the sequence (*motion estimation* or *fusion planning*)Reconstructing the HR image (*restoration*).

We will treat separately each of the three steps and how they have been implemented in this work.

### 2.1. HS Image Capturing: Acquisition

As briefly mentioned above, the process of acquisition can be defined as the combination of elements that enable to properly capture and store a sequence of HS images of the same scene with sub-pixel shifts between them, including the necessary processing techniques that makes it suitable for starting a process of super-resolution.

#### 2.1.1. HS Image Instrumentation

The most commonly used instruments to capture HS images are staring spectral arrays and push-broom scanners [34]. The first ones capture the whole scene at a time in a band-sequential format using a selectable filter. For this reason, staring spectral arrays have a limitation imposed by the low number of possible spectral filters that can fit into a single instrument. Furthermore, they are usually limited to the Visual and Near-Infrared (VNIR) spectral range, which together makes them less suitable for our target application. Instead, the push-broom scanners gather a complete spectrum of each point of a line at the same time, and generate the HS cube compiling the information obtained line-by-line. These sensors can expand towards the Short-Wave Infrared (SWIR) and they offer higher spectral resolution. A push-broom camera is employed in this work.

The three necessary subsystems in every HS acquisition system could be simplified as: lenses, image sensor and light source. The lenses focus the scene, the sensor records the HS data, and the light illuminates the scene. The optical system employed in our experiment is described in detail in [35], and consists of an HS camera coupled to a conventional light microscope. In consequence, the three aforementioned subsystems can be matched to our set-up as follows:The *lenses* subsystem in this experiment is a complex Olympus BX-53 microscope (Olympus, Tokyo, Japan) with a tube lens (U-TLU-IR, Olympus, Tokyo, Japan) that permits the attachment of a camera with a selectable light path and LMPLFLN family lenses (Olympus, Tokyo, Japan) with four different magnifications: 5×, 10×, 20× and 50×.The *image sensor* is a push-broom HS camera model Hyperspec^®^ VNIR A-Series from HeadWall Photonics (Fitchburg, MA, USA), which is based on an imaging spectrometer coupled to a Charge-Coupled Device (CCD) sensor, the Adimec-1000m (Adimec, Eindhoven, The Netherlands). This HS system works in the spectral range from 400 to 1000 nm (VNIR) with a spectral resolution of 2.8 nm, being able to sample 826 spectral channels and 1004 spatial pixels.The *light source* is embedded into the microscope and is based on a 12 V–100 W halogen lamp.

Figure 1 presents the actual acquisition system as described above.

As it is a push-broom system, a relative motion between the HS sensor and the targeted sample is needed in order to acquire HS data cubes. In our particular acquisition system, the HS sensor remains motionless while the sample to be scanned is moved taking advantage of the microscope motorized stage (Scanning Stage SCAN 130×85, Märzhäuser, Wetzlar, Germany). The movement in the X-Y directions has been automated, which can be controlled from the computer in order to synchronize the sample movement with the sensor acquisition process. Further details of the set-up can be found in [36].

As our camera has a pixel size of 7.4 μm, this results in a theoretical Field of View (FOV) of 1.5 mm for the 5× magnification, 750.45 μm for the 10× magnification, 375.23 μm for the 20× magnification and 148.59 μm for the 50× magnification. The mechanical resolution of the system is 0.01 μm and has an accuracy of 3 μm, which provides accurate movement of the specimens. Its relationship with the pixel size is a potentially critical parameter for our study, as it is the enabler for easy sub-pixel movement generation.

#### 2.1.2. HS Brain Histology Dataset Acquisition

By using the aforementioned HS system, several HS cubes were collected to create a SR dataset. The employed microscope histological samples were a micrometric ruler and brain cancer samples taken from slides such as the one presented in Figure 2a. Each sample area was captured 16 times at least, following a previously defined sub-pixel movement pattern, and composing a sequence. One of the used patterns is presented in Figure 2b. Sequences were captured at 20×, 10× and 5×, depending on the sample (which sequence corresponds to which magnification will be indicated later when presenting the results). Here, 20× is the highest magnification we are able to capture with this set-up due to the limitation imposed by the current light source. Each sequence has one ground truth image associated, with a magnification immediately further and centered in the same physical position than the first sample of the sequence.

#### 2.1.3. HS Data Pre-Processing

As is common to a push-broom acquisition architecture, there are several artifacts that can be generated in the image while it is being captured. For this reason, a pre-processing was deemed necessary to be applied on each captured image. Such pre-processing chain consists of capturing an image with the light source that will be used to illuminate the sample later, but without any sample in between. In this way, the complete light spectrum will enter into the HS sensor, generating the *White Reference* (WR) image. Consequently, the WR does not depend on the movement of the microscope platform. An example can be found in Figure 3.

The spatial resolution of these WR images is 100 × 1004 pixels for each band. Hence, there are 100 samples across each of the columns of the push-broom vertical scan. Such values will be averaged to obtain a 1 × 1004 × 826 HS image (of one single line) that will be used for the calibration of each frame of the captured HS image, independently. In the spectral signatures of Figure 3b it becomes evident that the light source is far from being uniform across the spectrum. This will impose a limitation on the signal-to-noise ratio of the bands in the upper and lower wavelengths. Furthermore, if we focus in wavelengths between 600 and 700 nm in Figure 3b, a variable absorption gap on some pixels can be appreciated. All these defects will affect negatively the final result of the acquisition, unless they are compensated.

Similarly, *Dark Reference* (DR) images need to be captured, with the camera shutter closed and the light off, to calibrate the noise of the sensor across the horizontal push-broom scan and all the bands. In consequence, the DR does not depend either on the movement of the microscope platform. An example of the DR images for the same band as before can be found in Figure 4.

The noise level measured along the spectra is random and uniform, in contrast to what was presented in the white reference case, and it has a similar Gaussian shape in all bands. An example of the measured Gaussian noise is presented in Figure 4b.

The 100 × 1004 × 826 dark reference images will again be averaged vertically to obtain the 1 × 1004 × 826 DR image (of one single line) that will be used in the calibration step as described in [36], and presented here in equation number 1. In-a-nutshell, the calibrated image (CI) is computed by subtracting the dark reference image from the raw image (RI), and normalizes the result over the white reference image (subtracting first the dark reference from it as well, as such background noise will be present also in the line-scan when capturing the white reference):(1)$$CI=\frac{RI-DR}{WR-DR}$$

We can see the result of a raw versus calibrated image on Figure 5, where it is evident that the vertical lines artifacts originated by the passes of the push-broom architecture are cleared out.

#### 2.1.4. HS Data Sequence Generation

Once the image is cleared of artifacts, two sequences will be generated through a motion vector matrix via average pooling 2 × 2 and 4 × 4 with the following method:1.Choose a point A, and starting from it, select a subset of the HS cube of size 256 × 256 in the spatial resolution plane, including all the corresponding spectral bands of the HS cube. Save this new HS cube as the reference image (Very-High Resolution Image—VHRI).2.Perform an average pooling 2 × 2 and 4 × 4 of the new HS cube. Save these new HS cubes as frame 0 of the High-Resolution Image (HRI) sequence and LR Image (LRI) sequence respectively.3.Choose a point B at 1-pixel distance from A and starting from it, select a subset of the HS cube of size 256 × 256 in the spatial resolution plane.4.Perform an average pooling 2 × 2 and 4 × 4 of the new HS cube. Save these new HS cubes as your frame 1 of the HRI sequence and LRI sequence respectively.5.Choose another point at 1-pixel distance from A, denoted C, and starting from it, select a subset of the HS cube of size 256 × 256 in the spatial resolution plane.6.Perform the same 2 × 2 and 4 × 4 pooling than in #2, and save it as frame 2 in the corresponding sequences.7.Perform the same pooling for each subset at 1-pixel and 2-pixels distance from A, including the results in the corresponding sequences.

Figure 6 aims to help visualising the process described above and understanding that a 1-pixel movement in one direction will create a half-pixel or quarter-pixel movement in the pooled images.

The resulting sequences will validate the proposed SR algorithm accuracy for fine motion estimation in a systematic manner, away from other factors as physical pixel shape of the sensor, or mechanical accuracy of the microscope stage/mechanical system. This will enable to apply the super-resolution algorithm to public image subsets such as Pavia University [37], and compare the performance with other state-of-art SR algorithms.

### 2.2. Motion Estimation: Fusion Planning

Motion estimation remains a critical process in video coding systems because it is computationally intensive. The algorithm under study is intended for implementation in a real-time system, and it is inspired on a fast implementation for video processing systems [38]: block-matching.

First, it can be trivially assumed that a two-dimensional motion estimator is accurate enough in our case of study, as the movements that will be seen are purposely designed to be in two dimensions only via the microscope X-Y scanning platform. The whole image will then be divided in smaller squared blocks, that are denoted as *macroblocks* (MBs). This process is performed with the purpose of reducing the memory demand of the program, as we know that we are in the case of global motion (the same motion vector will be applied to the whole image). Each MB will be individually evaluated using the Full Search Block Matching (FSBM) algorithm (or *exhaustive search* as in [39]) to find the possible motion vectors between the different frames. The position with minimal Sum of Absolute Differences (SAD) at the end of the process that iterates over a search area, will be assigned to the MB as a motion vector, first at pixel, then at half-pixel, and finally at quarter-pixel granularity. It is then when the FSBM algorithm finds an estimated motion vector that minimizes a displaced frame difference error [40].

In our case of study, we can validate the accuracy of the motion estimation because we have control over the sequence acquisition. Furthermore, we have the problem that the contrast across the different bands is highly different. Therefore, we must find a way to select the appropriate band or set of bands to perform such estimations, guaranteeing high-texture images for more accurate FSBM behavior. There are two potential approaches that can be selected here: (1) choosing the most suitable candidate among the bands to perform the motion estimation, or (2) combining the bands to compose an appropriate image that helps the matching algorithm to works properly. The first one has been tested on several histological brain images as well as non-medical images, and the result have not been easy to generalize. On the contrary, opting for the second branch has been proven to be a systematic method that directly benefits from the abundant spectral content of each scene. The algorithm that has been chosen to combine the bands to obtain higher-texture images is Principal Component Analysis (PCA) [41].

With the help of a PCA for the reference image, a combination of the different bands is proposed. Such combination is then applied to all the frames in the sequence, in a way that all of them will have an assigned synthetic band that accumulates high level of variance. Such extra variance will aid the block matching algorithm to estimate more accurately the global movement.

As can be clearly appreciated in Figure 7, there are bands that do not have enough contrast and seem almost unfocused. Hence, using these bands it will be very complicated to perform accurate motion estimation over the target sequence.

A measure of the contrast of an image can be appreciated in its histogram of light intensities. The wider the histogram, the higher the contrast in that image. On the contrary, a narrow histogram would indicate that the intensities are very similar to one another across the whole image. Hence, it will be more difficult to minimize the SAD function, which is the core of the Block Matching algorithm (BM) for accurate motion estimation. The result of the PCA in the image presents a clear border profile on the shapes inside the brain tissue sample, which will aid the BM algorithm to identify MB movements.

In all the HS images that have been analyzed, it was straightforward to obtain a single channel PCA image that accumulates at least 65% of the variance of the whole HS image using MATLAB^®^ PCA function. This was experienced to be enough for sub-pixel motion estimation. It is taken as an assumption in this study that an artificial single-band representation which accumulates at least 65% of the total variance can be elaborated for any target sequence on the pathology field.

The end result of the motion estimation and fusion planning phase of the algorithm is a labeled sequence of LR frames. The label contains, for each frame, the relative motion that has been estimated with respect to the reference frame.

### 2.3. Image Reconstruction: Restoration

As was explained in the introduction, the approach used for SR in this study is the construction of HR HS images through the combination of several LR images with spectral information in the same bands and sub-pixel displacement among them, which will be denote as *frames*. Several frames compose what we denote as *sequence*. The reference algorithm that will be used for comparing image quality improvement is the Bilinear Interpolator. This is justified because it is a simple and powerful algorithm, providing a well-balanced trade-off between runtime and quality, which is what we are looking for. Secondly, it is a common comparison reference point for evaluating the performance of super-resolution algorithms [42]. Details of how to implement the BI can be found in [33].

The original SR algorithm comes from a previous study on applied video processing techniques by [43,44], which purpose was to achieve better image quality on a real-time application. The approach relies on considering each LR frame as a down-sampled version of a higher-resolution image, which will ultimately be used as the true scene. The SR algorithm will align LR observations of the same sequence with sub-pixel accuracy, combining them into a HR grid. This is only valid when an accurate relative motion estimation has been previously achieved, as it was explained in Section 2.2. The overall approach is based on applying SR to monochromatic frames, fundamentally into the luminance, and then interpolating the chrominances. A general diagram that depicts the process can be found in Figure 8.

The proposed SR algorithm used for our study will opt for applying the same technique used in the luminance, but individually to each band of the HS image. The algorithm works as follows: once the motion estimator has computed the set of motion vectors for each macroblock (MB) of the sequence, those vectors will be used to shift every MB in a higher resolution grid. The algorithm considers several parameters of the motion estimation and the surrounding pixels to weight the importance in the grid of each sub-pixel coming from different frames, calculating a final value for the sub-pixel position using relatively simple mathematical function. This process has been denoted as *Shift and Add*. Finally, if there is any pixel which has been not filled throughout this process, it will be marked as *hole*, and then interpolated using a bilinear surface interpolator. The same strategy will be applied to each band using the same motion vector matrix that was calculated in the artificial single-band representation calculated via PCA.

### 2.4. Evaluation Metrics

Three different metrics have been considered to analyze the quality of the super-resolved HS images obtained by the employed algorithms:Structural Similarity Index ([45]) is a full-reference metric which measures the image degradation as perceived change in structural information. Higher values mean better image quality, and it is calculated as follows:
(2)$$SSIM\left(x,y\right)=\frac{\left(2\mu_{x}\mu_{y}+C_{1}\right)+\left(2\sigma_{xy}+C_{2}\right)}{\left(\mu_{x}^{2}+\mu_{y}^{2}+C_{1}\right)\left(\sigma_{x}^{2}+\sigma_{y}^{2}+C_{2}\right)},$$
where $\mu_{x}$ is the average of *x*, $\mu_{y}$ is the average of *y*, $\sigma_{x}$ is the variance of *x*, $\sigma_{y}$ is the variance of *y*, $\sigma_{xy}$ is the covariance between *x* and *y*, $C_{1}={\left(k_{1}L\right)}^{2}$ and $C_{2}={\left(k_{2}L\right)}^{2}$ are two constants to stabilise the division with weak denominator, $k_{1}=0.01$, $k_{2}=0.03$ and *L* is the dynamic range of the pixel values.Peak Signal-to-Noise Ratio (PSNR) is an absolute error metric that measures the relationship between the maximum possible power of a signal and the power of corrupting noise that affects the fidelity of its representation. Higher values mean better image quality, and it is calculated as follows:
(3)$$PSNR=10\log_{10}\left(\frac{R^{2}}{MSE}\right),$$
where *R* is the maximum fluctuation in the input image data type, and *MSE* the Minimum Square Error.Spectral Angle Mapper (SAM, [46]) is a full-reference metric which measures the spectral degradation of a pixel with respect to a reference spectral signature, in the form of an angle between their two vector representations. Values closer to zero mean better image quality, and it is calculated as follows:
(4)$$SAM\left(x_{n},{\hat{x}}_{n}\right)=arccos\left(\frac{\langle x_{n},{\hat{x}}_{n}\rangle }{∥x_{n}{∥}_{2}{∥{\hat{x}}_{n}∥}_{2}}\right),$$
where $x_{n}$ and ${\hat{x}}_{n}$ are the individual pixel spectral vectors of the reference and super-resolved HSI respectively, and SAM is the average SAM of all pixels in the HS cube.

The escalation factor (EF) denotes the spatial resolution enhancing objective of the algorithm. Results for ×2 and ×4 have been obtained in all the image sequences (128 × 128 pixels images were super-resolved to 256 × 256 and 512 × 512 pixels per band). *N* denotes the number of frames that have been combined in the sequence under study.

Additionally, two different scores are proposed for understanding the performance of the proposed SR algorithm with respect to other methods found in the literature:(5)$$Score1=\frac{PSNR}{SAM},$$
where PSNR and SAM are defined in Equations (Equation 3) and (Equation 4) respectively.
(6)$$\begin{array}{ccc} Score2= & \frac{PSNR}{SAM\times Runtime^{*}}=\; & \frac{Score1}{CF_{CPU}\times CF_{volume}\times Runtime}, \end{array}$$
where *Runtime^*^* is the corrected runtime by factors $CF_{CPU}$ and $CF_{volume}$, properly described in Section 3.1.2; PSNR and SAM are defined in Equations (Equation 3) and (Equation 4) respectively.

As it can be deducted from examining Equations (Equation 5) and (Equation 6), the two proposed scores have the following properties:The higher the score, the better the algorithm.Metrics that have *infinity* as ideal value are in direct proportion with both scores.Metrics that have *zero* as ideal value are in inverse proportion with both scores.

### 2.5. Processing Platform

All the results that will be presented in Section 3 have been obtained with a commercial computer with a CPU Intel^®^ Core^TM^ i7-3540M running at 3.0 GHz and 8 Gigabytes of DDR3 RAM clocked at 1600 MHz. The MATLAB^®^ version used was 2020a, and the algorithm was implemented in C language.

## 3. Results

To be able to understand the performance of the algorithm, a set of five HS sequences were studied. First, a popular public HS image was run through the algorithm: Pavia University [37]. This served us to compare the results with other state-of-the-art algorithms and to generalize the relevance of the algorithm to different sensors and data applications. The other four HS sequences were generated from histology samples of brain tissue acquired using the considered HS microscopic instrumentation. For all histology samples, the algorithm was run for EF of 2 and 4, and always combining the 25 images that are available inside the sequence. The combined results can be inspected in Figure 12 together with a BI of the same respective factor, for the reader to be able to visually self-assess image quality improvement.

### 3.1. Sequence 1—Pavia University

The first sequence under study has been taken from a public dataset: Pavia University [37].

The sequence was generated using the method presented in Section 2.1, applied to the HS image dataset from the public repository of the ROSIS-3 (Reflective Optics System Imaging Spectrometer) over the city of Pavia (Italy), in particular the image that captures the University campus. The spatial resolution of the image is 610 × 340 pixels, having 115 spectral bands in the range from 430 to 860 nm, with a bandwidth of 4.0 nm.

#### 3.1.1. Proposed SR Algorithm vs. Interpolation

The proposed SR algorithm has been run for EF of 2 and 4, combining nine images that are available inside the sequence (reference frame plus eight extra global movements). The results for EF = 2 and EF = 4 can be seen in Figure 9, together with a BI of the reference image of the same factor, for the reader to be able to self-assess the image quality improvement. The displayed band corresponds to a 630 nm wavelength.

Using EF = 2 (Figure 9a) it can be appreciated more vividly the parking lot and the white structure on the right-hand side of the rounded structure, that the sharpness of the different elements in the images has improved. However, using EF = 4 (Figure 9b) it is more evident that the interpolated image has difficulties following diagonal straight lines, producing a clear square effect in the University building, in opposition to the super-resolved image. The parking lot and the zoomed buildings present a clear improvement with the technique under study.

Additionally, to quantitatively assess the improvement of the resulting image using the proposed SR algorithm with respect to the BI across the acquired spectra, the average SAM has been calculated for both images with respect to the reference image. Results are presented in Table 1.

Instead of assessing the capability to maintain spectral similarities, now each individual band has been evaluated against its corresponding interpolated counterpart using PSNR and SSIM metrics. Results are presented in Figure 10.

In EF = 2 (Figure 10a) and EF = 4 (Figure 10b), the values indicate a consistent improvement along the studied spectra for both SSIM and PSNR. For the first one, average values of 8.69% and 0.78 dB improvement in SSIM and PSNR respectively, with more stable performance between 450 and 800 nm wavelengths with respect to the previous sequences. A clear positive *protuberance* over average in PSNR can be noticed in the ranges from 550 to 700 nm approximately, while in SSIM the *protuberance* is slightly negative, approximately 1% less than average.

However, in EF = 4, SSIM improvement is now around 15% average, and the PSNR improvement is nonexistent anywhere but for the same wavelength range 450 and 700 nm (roughly the same as the previously described *protuberance* for EF = 2).

For the sake of validation of the proposed approach, another verification has been carried out with the Pavia University dataset: effective representation of what is the performance increase of the algorithm when increasing the number of images (N) combined during the super-resolution process. Results are presented on Figure 11, where the performance of the SR algorithm increases with larger number of images involved in the SR algorithm, as intuitively expected. In other words, the fewer the images, the more similar to the interpolation it behaves. This is reasonable as less images mean that there will be less weighted values in the very high resolution grid, and the holes are effectively filled with interpolated values.

After the integration of eight images, the performance is not improved using higher number of N. This is produced because the movement in the sequence has been generated artificially. In practice this means that the frames have been purposely created to fill the most important holes of the image in the less number of images. The rest of the information is effectively superfluous for improving SSIM metric.

#### 3.1.2. Proposed SR Algorithm vs. State-of-the-Art Algorithms

Thanks to the use of Pavia University dataset, it was possible to compare the algorithm under study with other state-of-the-art solutions. As execution time (or *runtime*) is an important parameter to our purpose in this article, it was necessary to make sure that the execution times of other algorithms are compared in equivalent conditions. Therefore, two correction factors have been used:Using the benchmarks in [47], it has been deducted that the CPU in [31] is 1.59 times faster for single-core use than the computer used for the results of this article. In consequence, 1.59 will be used as a correction factor for the comparison of the execution times, and will be denoted as $CF_{CPU}$.Reading in detail the Pavia University subset used in [31], it can be appreciated that the volume of the HS cubes handled is 1.28 smaller than our own. Hence, it was considered appropriate to use 1.28 as correction factor for the processing time presented there, and will be denoted as $CF_{volume}$.

Table 2 presents the comparison data and highlights in bold the best performance on each category. The SR algorithm proposed in this article presents the second best SAM, and the second best PSNR, hence clearly remains second in *Score 1*. Not considering the runtime, it is clearly outperformed by the LRSR HSI-MSI algorithm, although well above the other four candidates. Unfortunately, due to the unavailability of the code for the other candidate algorithms, comparison of SSIM could not be obtained.

When introducing the corrected runtime in the equation, the proposed algorithm outperforms. Even if the algorithms in [30] do not give runtime execution information, for their structure it is expected that they should be similar to HySure or STEREO, remaining FUSE the only one that can compete with our algorithm on this metric.

### 3.2. Sequence 2—High-Density Brain Tissue

The first brain sequence under study was taken from a dense brain tissue, and created with the method described in Section 2.1. The selected band displayed in Figure 12a,b corresponds to 621 nm.

As it can be appreciated on both cases, the interpolated image presents a more squared resolution of all the organic rounded details, as well as evident signs of overall lower quality to the naked human eye.

In order to assess quantitatively the improvement of the resulting image of the algorithm with respect to the BI across the acquired spectra, the average SAM has been calculated for both images with respect to the reference image. The results for EF = 2 and 4 are presented in Table 3.

Instead of assessing now the capability to maintain the spectral signature (as carried out through SAM), now each individual band has been evaluated against its corresponding interpolated counterpart through PSNR and SSIM. The results are presented in Figure 13.

In EF = 2 (Figure 13a) the values indicate a consistent improvement along the studied spectra for both SSIM and PSNR, with average values of 6.3% and 0.89 dB improvement respectively. A clear negative protuberance on the range of 430 to 550 nm wavelengths can be noted on both PSNR and SSIM metrics. In EF = 4 (Figure 13b), the values indicate a much higher improvement on SSIM with respect to EF = 2, while a more modest improvement on the PSNR side. Average values indicate 20.22% and 0.05 dB improvement respectively. The previous negative protuberance on SSIM for EF = 2 does not appear any more, and the improvement monotonically increases with frequency. PSNR manifests a decrease of performance down to zero improvement with respect to the BI in the bands around the 715 nm wavelenth.

### 3.3. Sequence 3—High Background Content Brain Tissue

The third sequence under study has been taken from a brain tissue sample with high background content, and also created with the method described in Section 2.1. The selected band (number 304) displayed in Figure 12c,d corresponds to 621 nm.

As it can be appreciated, for EF = 2 (Figure 12c), the super-resolved image is able to capture the background texture, and presents signs of overall better quality than the interpolated image already to the naked human eye, despite the fact that the algorithm has visually provoked a generalised whitening of the image. This kind of effect can be easily corrected through re-calibration, although it was evidently undesired and should be further investigated for improving the algorithm. For EF = 4 (Figure 12d), the interpolated image presents again a more squared resolution of all the organic rounded details than the super-resolved image. Again, the same whitening effect is present than for the EF = 2 case.

In order to assess quantitatively the improvement of the proposed algorithm’s outcome image, the average SAM has been calculated for both images with respect to the reference image. The results for EF = 2 and 4 are presented in Table 3.

Similarly to what was performed in previous sequences, each individual band has been evaluated against its corresponding interpolated counterpart through PSNR and SSIM. The results are presented in Figure 13.

For EF = 2 (Figure 13c), the values indicate a consistent improvement along the studied spectra for both SSIM and PSNR, with average values of 11.6% and 0.89 dB improvement respectively. It is interesting to appreciate the inverse proportionality of the SSIM versus the PSNR curve: when one is decreasing with respect to the BI, the other increases, and vice-versa, remaining both flat in similar band ranges. For EF = 4 (Figure 13d), the values indicate also consistent improvement along the studied spectra, with a much higher improvement in terms of SSIM overall. Average values indicate 19.30% and 0.11 dB improvement respectively. In terms of trend, similarly to what happens on the EF = 2 case, there is certain inversion of the growth on SSIM and PSNR curves.

### 3.4. Sequence 4—Brain Tissue with Small Objects

The fourth sequence under study has been taken from a brain tissue sample with a lot of small objects floating in the background texture, and also created with the method described in Section 2.1. The selected band (number 364) displayed in Figure 12e,f corresponds to 592 nm.

As can be appreciated, for EF = 2 (Figure 12e), the interpolated image presents a more squared-shaped resolution of all the organic rounded details, and signs of overall lower quality to the naked human eye, but this time is less noticeable, so it is even more important to go to the quantitative metrics to understand the performance.

Moreover, with EF = 4 (Figure 12f), it is more evident that the interpolated image presents again a more squared resolution of all the organic rounded details, as well as evident signs of overall lower quality to the naked human eye.

As was proposed in the other sequences, in order to assess quantitatively the improvement of the algorithm’s resulting image with respect to the linear interpolation across the acquired spectra, the average SAM has been calculated for both images with respect to the reference image. The results for EF = 2 and 4 are presented in Table 3.

Similarly to what was performed in previous sequences, each individual band has been evaluated against its corresponding interpolated counterpart through PSNR and SSIM. The results are presented in Figure 13.

For EF = 2 (Figure 13e), the values indicate a consistent improvement along the studied spectra for both SSIM and PSNR, with average values of 23.8% and 0.73 dB improvement respectively, and a more stable performance between 500 and 900 nm wavelengths with respect to the previous sequences. For EF = 4 (Figure 13f), the values indicate improvement along the studied spectra, with a much higher increment on terms of SSIM overall, while the PSNR falls now to slightly negative values for the 500 to 800 nm wavelengths. Average values indicate 33.83% and 0.15 dB improvement respectively.

### 3.5. Sequence 5—Highly-Granular Brain Tissue

The fourth and last brain sequence under study has been taken from a sample where a great number of individual cells are present at different depths, and also created with the method described in Section 2.1. The selected band (number 232) displayed in Figure 12g,h corresponds to 569 nm.

For EF = 2 (Figure 12g), it can be appreciated that the more square-shaped pixel resolution together with the granularity produces an effect of vertical lines in the zoomed rectangle of the interpolated image. With EF = 4 (Figure 12h), it is now more evident that the interpolated image presents again a more square-shaped resolution of all the organic rounded details, as well as evident signs of overall lower quality to the naked human eye.

As was proposed in the other sequences, in order to assess quantitatively the improvement of the algorithm’s resulting image with respect to the linear interpolation across the acquired spectra, the average SAM has been calculated for both images with respect to the reference image. The results for EF = 2 and 4 are presented in Table 3.

Similarly to what was performed in previous sequences, each individual band has been evaluated against its corresponding interpolated counterpart through PSNR and SSIM. The results are presented in Figure 13.

For EF = 2 (Figure 13g), the values indicate a consistent improvement along the studied spectra for both SSIM and PSNR, with average improvement values of 8.69% and 1.04 dB respectively. A more stable performance can be noted between 500 and 900 nm wavelengths with respect to the previous sequences. The trends of PSNR and SSIM are again opposite along the spectra (SSIM grows when PSNR decreases and vice-versa). For EF = 4 (Figure 13h), the values indicate a much higher improvement in terms of SSIM overall, while a much more modest improvement on the PSNR side with respect to EF = 2. Average values indicate 26.32% and 0.26 dB improvement respectively. It is also relevant that SSIM increment grows monotonically with longer wavelengths.

## 4. Discussion

A comparison summary of all the presented results has been displayed in Figure 14. Thanks to the calibration process of the HS cubes prior to the fusion, the effect of the non-uniform spectral profile of the microscope light has been de-correlated to a big extent. Hence, it cannot be anticipated in the vast majority of the studied bands.

It is sufficiently evident that the SSIM metric is drastically decreased when the noise level is high, a usual circumstance in the bands of the 400–500 nm and 700–1000 nm range. This causes the U-shaped SSIM curves for all the images and explains also a flatter response (in terms of improvement) in the central bands (500–700 nm).

On the contrary, the PSNR metric does generally give a more stable and flat performance improvement across the spectra. It is noticed that EF = 2 gives very similar improvement results for the five study cases (0.6 to 1.2 dB improvement). Although it is expected the same for EF = 4, it is evident from Figure 14d that the algorithm’s capability to significantly improve a BI should only be awarded for tissues with high granularity. Looking at the images, this can be intuitively explained because the size if the grains in *sequence 5* provokes that they are mixed in a non-linear manner during the pooling. Therefore, trying to recover the scene through linear interpolation is not effective; instead, the SR algorithm under study manages to combine the information from the different frames in the sequence in a smarter non-linear way.

Studying the Pavia University image together with the histology samples demonstrates that images with radically different nature and registration, unfortunately, achieve very different results in terms of performance of the algorithm. This effect is not only appreciated in the analysed spectra, but also in the whole HS cube.

The *protuberance* seen in the Pavia University sequence is remarkable; there is nothing similar in the other sequences, and it overlaps with the visible spectra. It does seem reasonable that, as we encounter the highest intensity of solar illumination coinciding with the visible spectra, the sensor is more able to detect sharper details. On the one hand, noise will be mostly white, and integrating several images would help to cancel such noise and improve the PSNR. On the other hand, using the illumination system of the HS microscope based on halogen light, we do not perceive such a *bump*.

Regarding the SAM metrics, it does not seem to be a clear relationship between the SAM values at EF = 2 and at EF = 4 in the same image, nor between different images. Nevertheless, it is remarkable that the proposed algorithm for SR has good spectral performance despite its low execution time and complexity.

Considering the reduced time executions achieved (few milliseconds per band), it is definitely possible to study the potential real-time execution of the algorithm with a dedicated processing system and some optimisations. The most relevant improvement would be to correlate the global motion estimation directly with the mechanical movement of the microscope, de-scoping the block-matching function, which is today measured to be 12–15% of the execution time of the super-resolution function.

## 5. Conclusions

An HS image SR algorithm for motion controlled environments has been presented. It has been demonstrated through the Pavia University dataset that it can outperform similar state-of-the-art algorithms. Its low execution time compared to other methods offers the possibility to study its use in real-time applications. The algorithm has demonstrated to consistently improve the image quality of the sequences with respect to a BI. Nevertheless, the performance in the selected metrics varies significantly between sequences. The algorithm has also demonstrated to be an efficient noise-filter in the noisiest bands of the HS cubes analysed, being able to improve the spectral signature fidelity of the pixels through the reconstruction process. This last feature, and the fact that the algorithm has demonstrated dealing with real medical images across the whole spectra of the sensor (400–1000 nm), enables its use in the field of medical histology, and paves the way to its future application to improved tissue classification.

## Figures and Tables

**Figure 1 sensors-23-01863-f001:**
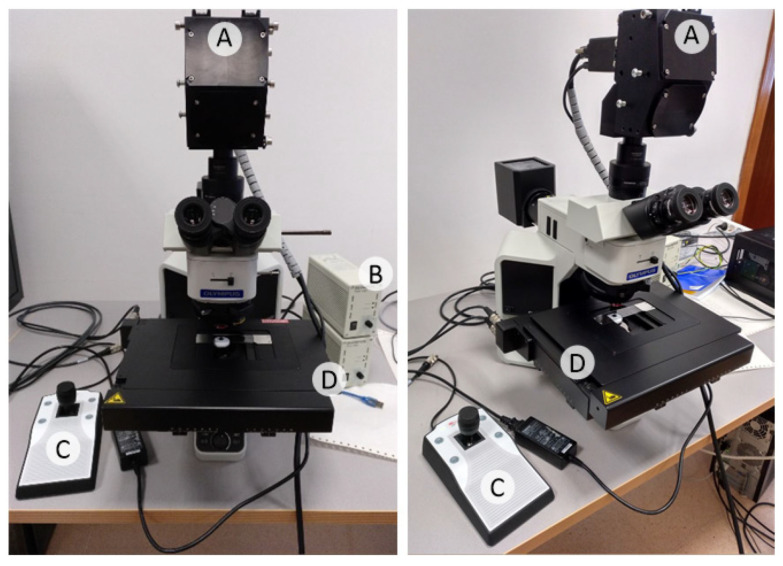
Microscopic HS acq. system. (A) HS camera. (B) Light source. (C) Positioning joystick. (D) Motorized stage.

**Figure 2 sensors-23-01863-f002:**
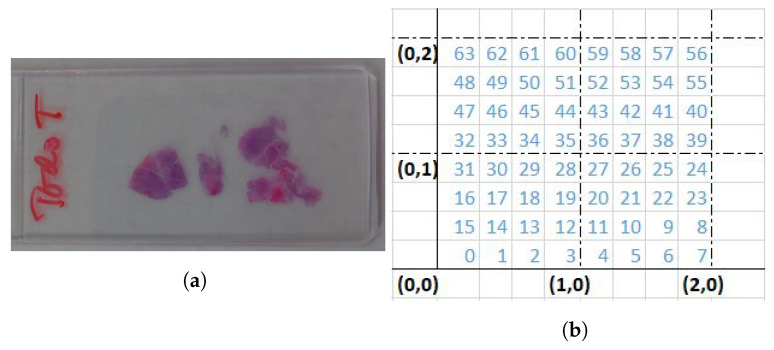
HS image dataset: (**a**) Brain histology sample, (**b**) labeling of 1/4-pixel movements around start coordinates (0,0).

**Figure 3 sensors-23-01863-f003:**
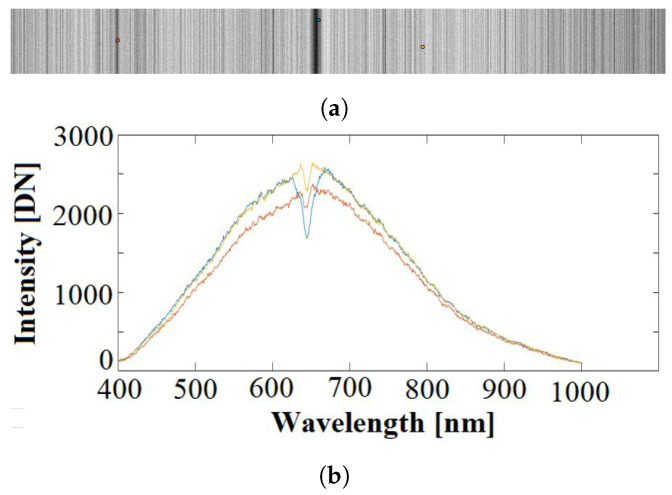
Example WR image for light source calibration: (**a**) Band number 319 (361 nm) where three pixels have been marked in red, blue and yellow. (**b**) Spectral signatures of the three marked pixels (Intensity is presented in sensor units).

**Figure 4 sensors-23-01863-f004:**
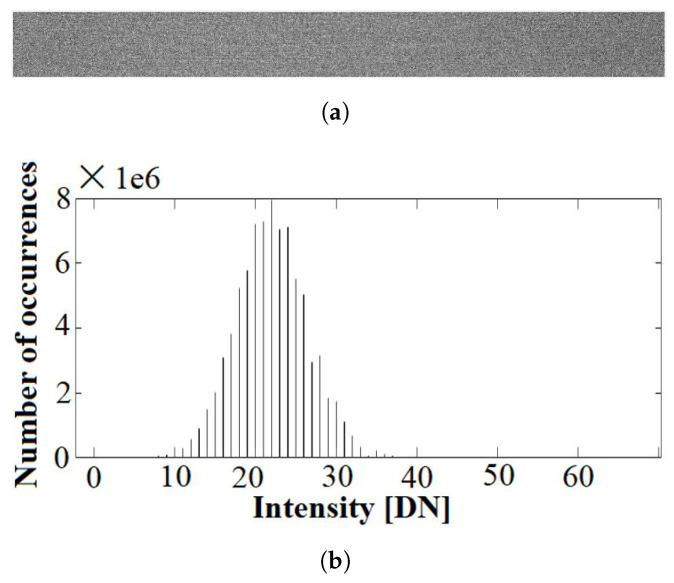
Example DR image for background noise calibration: (**a**) Band number 319 (which corresponds to 631 nm). (**b**) Histogram of the noise intensity when shutter is closed.

**Figure 5 sensors-23-01863-f005:**
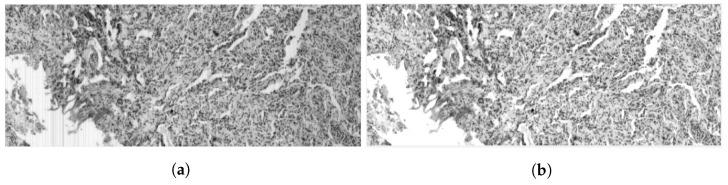
Raw image (**a**) and calibrated image (**b**) for a brain tissue histology sample at band number 302 (619 nm).

**Figure 6 sensors-23-01863-f006:**
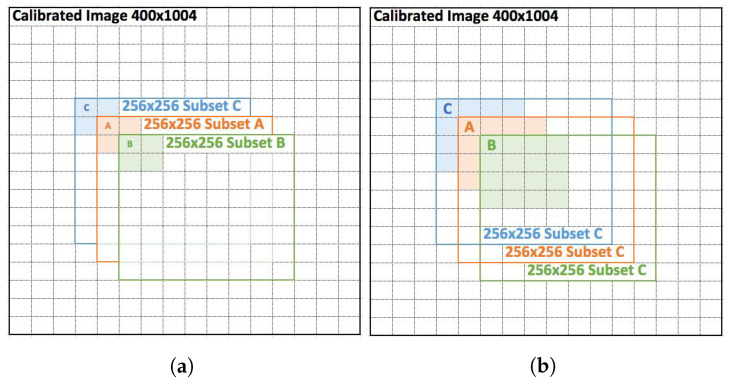
Illustration of the sequence generation. On (**a**) colored areas represent the target for 2 × 2 average pooling on each of the subsets that will create the 2D sub-pixel motion in the HR image sequence. On (**b**) colored areas represent the target 4 × 4 average pooling instead. Sequence B has a motion vector (1,1) with respect to A, and (2,2) with respect to C.

**Figure 7 sensors-23-01863-f007:**
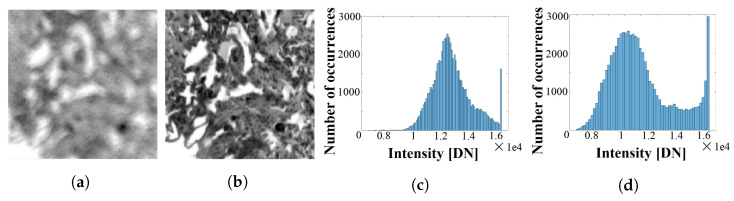
Brain histology sample from our database: (**a**) band 321 (633 nm), blurry and difficult to be used for sub-pixel motion estimation, with (**c**) its corresponding histogram of intensities; (**b**) obtained image from PCA that will be used for motion estimation, and (**d**) its associated histogram.

**Figure 8 sensors-23-01863-f008:**
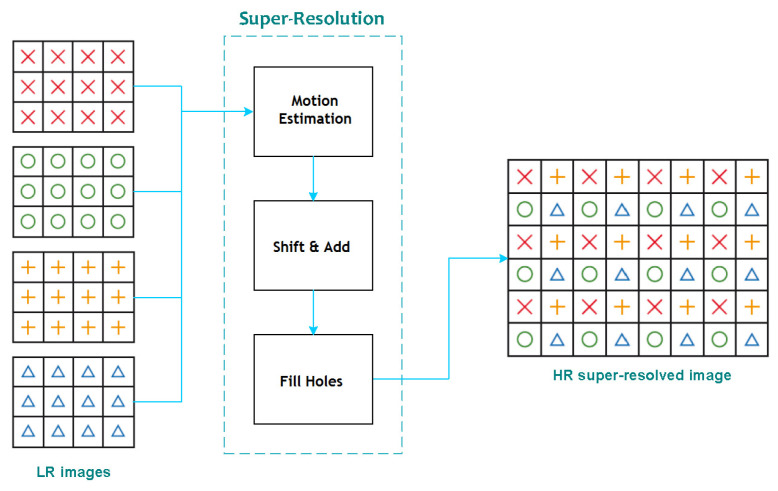
General diagram of the super-resolution approach [10].

**Figure 9 sensors-23-01863-f009:**
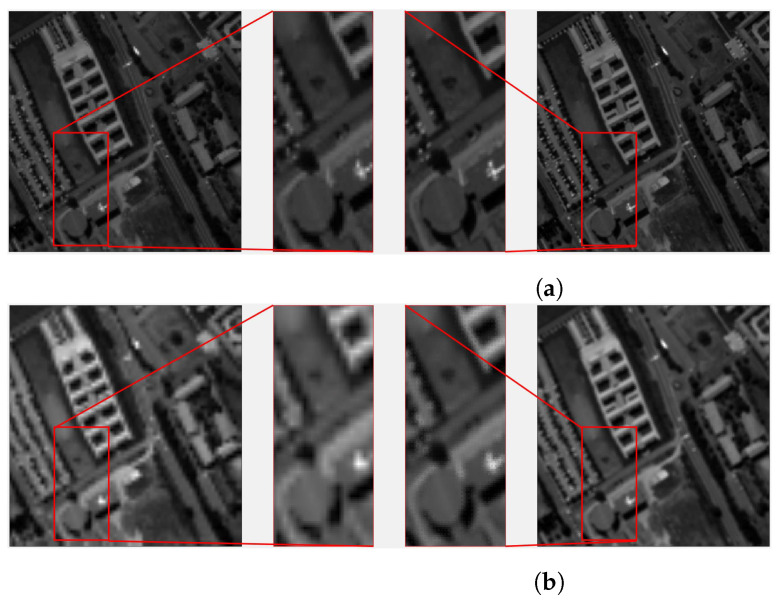
Visual quality inspection for *Sequence 1*, (**a**) EF = 2 (N = 8) and (**b**) EF = 4 (N = 24) at 630 nm band. On the left, the interpolated image and with a zoom over the area of interest; on the right, the super-resolved image with its zoom.

**Figure 10 sensors-23-01863-f010:**
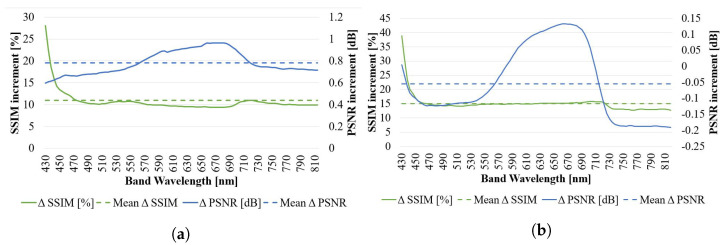
Performance variation for the proposed SR algorithm across the spectrum for PSNR and SSIM metrics in *Sequence 1*: (**a**) EF = 2 (N = 8) and (**b**) EF = 4 (N = 24). Average values are presented in dashed lines.

**Figure 11 sensors-23-01863-f011:**
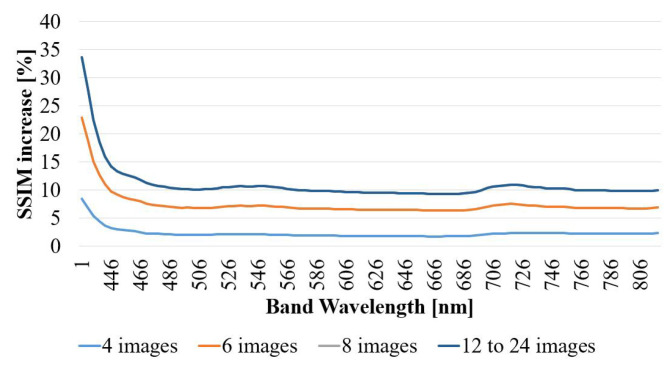
Performance variation of the algorithm for SSIM metric, varying the number of integrated images (N). Performances for N = 8, 12, 16, 20 and 24 are superposed in the figure for being extremely similar. *Sequence 1*, EF = 2.

**Figure 12 sensors-23-01863-f012:**
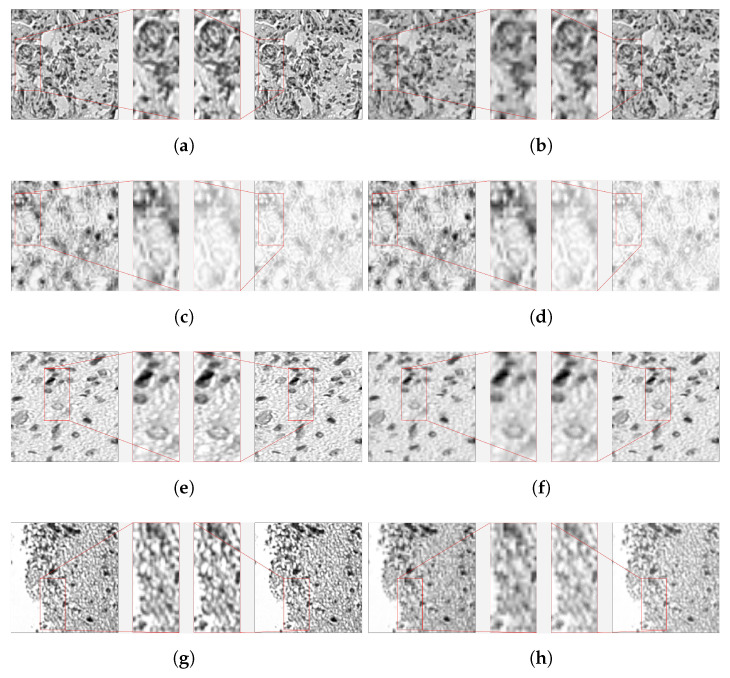
Visual quality inspection for *Sequences 2* to *5*, and N = 25. The **interpolated image** and its corresponding zoom can be found on the *left-hand side*; instead, *on the right*, the **super-resolved** image with its zoom. (**a**,**c**,**e**,**g**) capture sequences 2, 3, 4, 5 respectively for EF = 2. Results for EF = 4 is presented in (**b**,**d**,**f**,**h**), at their right side.

**Figure 13 sensors-23-01863-f013:**
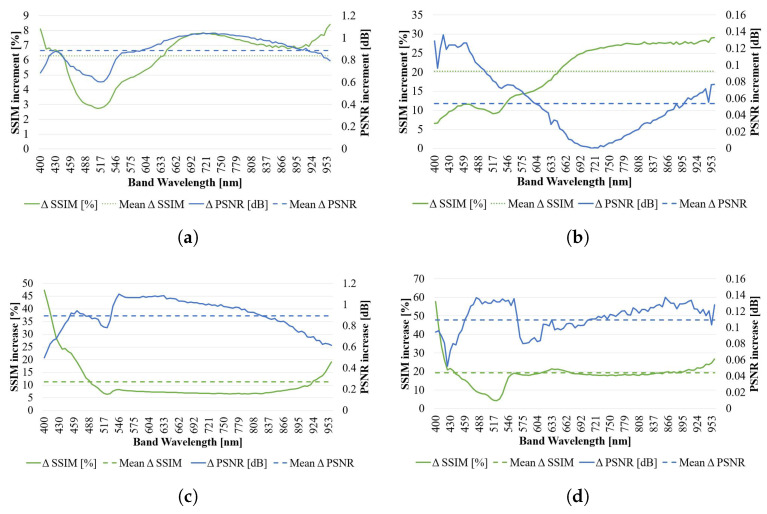
Variation of the performance of the proposed SR algorithm across the spectrum for PSNR and SSIM metrics in *Sequences 2* to *5*. All sequences consider N = 25. (**a**,**c**,**e**,**g**) capture sequences 2, 3, 4, 5 respectively for EF = 2. Their corresponding result for EF = 4 is presented in (**b**,**d**,**f**,**h**), all at their immediate right side. Average values for both performances along the studied spectra are presented in dashed lines as a general indication.

**Figure 14 sensors-23-01863-f014:**
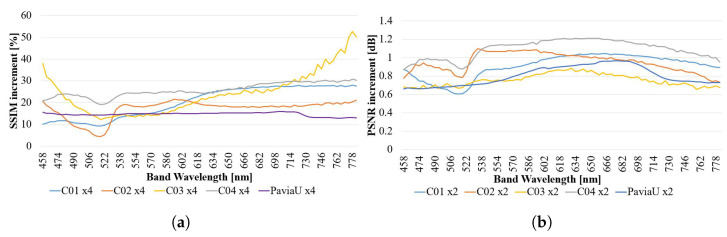
Variation of the performance of the algorithm across the spectrum for the different sequences: (**a**) SSIM for EF = 2 and N = 25, (**b**) PSNR for EF = 2 and N = 25, (**c**) SSIM for EF = 4 and N = 25, (**d**) PSNR for EF = 4 and N = 25.

**Table 1 sensors-23-01863-t001:** Quantitative assessment results of the proposed SR algorithm versus interpolation for *Sequence 1*. ↑ *SAM* represents the gain of the proposed algorithm in SAM with respect to a BI. Five executions of the same program have been carried out to confirm the execution time.

EF	N		SAM [deg]	↑ SAM [%]	Execution Time [s]
2	8	Interpolated	4.272	11.68	0.072±0.01
		Proposed	3.822		2.49±0.1
4	24	Interpolated	7.026	4.36	0.145±0.01
		Proposed	6.732		4.60±0.1

**Table 2 sensors-23-01863-t002:** Performance comparison of different SR algorithms on the Pavia University dataset. In **bold** the winner on each metric. *Score 1* is defined in Equation (Equation 5) and *Score 2* in Equation (Equation 6).

Algorithm/Metrics	SAM (°)	PSNR (dB)	Score 1	Runtime (s)	Score 2
*Ideal value*	0	*∞*	*∞*	0	*∞*
STEREO [31]	4.55	22.50	4.86	26.4	0.0920
FUSE [31]	5.54	21.09	3.81	**0.5**	3.7437
HySure [31]	4.81	21.18	4.40	82.5	0.0262
LRSR HSI-PAN [30]	4.56	33.69	7.39	-	-
LRSR HSI-MSI [30]	**1.81**	**43.89**	**24.24**	-	-
*Proposed*	*3.82*	*36.84*	*9.69*	*1.67*	* **5.8052** *

**Table 3 sensors-23-01863-t003:** Quantitative assessment results of the proposed SR algorithm versus interpolation for sequences 2 to 5. *↑SAM* represents the gain of the proposed algorithm in SAM with respect to a bi-linear interpolation. Five executions of the same program have been carried out to confirm the duration.

			SAM vs.	↑SAM	Execution Time
			Reference [deg]	[%]	per Band [ms/band]
**Sequence 2**	* **EF = 2** *	Interpolated	2.398	9.01	20±1
	* **N = 25** *	Proposed	2.200		226±21
	* **EF = 4** *	Interpolated	4.641	0.00	19±1
	* **N = 25** *	Proposed	4.641		205±5
**Sequence 3**	* **EF = 2** *	Interpolated	3.209	7.91	19±1
	* **N = 25** *	Proposed	2.974		245±19
	* **EF = 4** *	Interpolated	4.700	10.00	20±1
	* **N = 25** *	Proposed	4.272		201±5
**Sequence 4**	* **EF = 2** *	Interpolated	3.601	7.26	19±1
	* **N = 25** *	Proposed	3.358		245±21
	* **EF = 4** *	Interpolated	4.178	1.28	19±1
	* **N = 25** *	Proposed	4.125		191±9
**Sequence 5**	* **EF = 2** *	Interpolated	2.278	9.84	19±1
	* **N = 25** *	Proposed	2.074		264±20
	* **EF = 4** *	Interpolated	8.021	0.00	19±1
	* **N = 25** *	Proposed	8.021		201±4

## Data Availability

All the image sequence are available in a private server of the IUMA, please, contact curbina@iuma.ulpgc.es for obtaining a private download link; no charges will be applied for the download. MDPI Research Data Policies are available at https://www.mdpi.com/ethics, accessed on 2 February 2023.

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
