# Peer review of "Towards Real-Time Hyperspectral Multi-Image Super-Resolution Reconstruction Applied to Histological Samples"

_sensors, 2023, doi:10.3390/s23041863_

Round 1

Reviewer 1 Report

 15 January 2023

Reviewer’s comments on manuscript „Towards real-time hyperspectral multi-image super-resolution reconstruction applied to histological samples (Sensors#2165124)” by C. Urbina Ortega et al.

This is a well-written manuscript about developing a hyper-spectral multi-imager microscopic system. Since these systems play an increasing role in almost every aspect of the every-day life spanning very diverse areas such as food quality checking, remote sensing from space, and medical histological analyses, this subject is important and actual. The present manuscript aims at the biomedical applications of the developed new ideas for acheaving super-resolution. The work has been carried out precisely and the organization of the manuscript regarding the descriptions of the applied methods and results is also in place. Weakness is the English style. My comments below mainly refer to the better understanding of the subject.               

Subjectal concerns:

1.      In the „Introduction” the fundamental role of the „sub-pixel motions” in determining super-resolution has been mentioned. Because of its importance in the judgement of the whole procedure, this deserves a little more attention. It seems as if the larger resolution would be thanked to a some kind of a „noise benefit”, the central notion of something in the field of „non-linear dynamics” what is known as „stochastic resonance (SR)”. Please mention the similarities and differences in the „Introduction”, if it is relevant. The „sub-pixel motion” can also be random? If the „subpixel-motion” can be random, what is its optimal spectrum for the largest resolution enhancement?

2.      The notion of the „escalation factor (EF)” would deserve also a little more attention. This is the enhancement in resolution caused by the above „sub-pixel motions”. The resolution enhancement in STORM and STED microscopies are characterized by extentions of the conventional Abbe-formula for microscope resolution. Is the EF factor calculated also by such a formula? If there exists such a formula, please indicate it in the text. Please, extend the text on EF with some sentences describing the role of this number.

3.      As a control method, „bilinear interpolation” is applied. Please give a little description of this approach in the „Materials and Methods”, with mathematics if it applies.              

Formal concerns:

1.      In Line 411: „from” is enough.

2.      In Line 420: correcrly „display”.

3.      In Figures, the letters for identifying panels are badly seen, please correct them, in all figures.

4.      In Legend for Figure 12, correctly is „The”.

5.      In Lines 512 and 517 please write „than” instead of „that”.

6.      In Line 518, correctly is „ … the whitening effect, the same as that for the EF=2 case, is …”.

7.      In Line 529, please give the definition of „BI”.

8.      In Line 591: correctly is „ … it can not be anticipated …”. Alternatively: „ … it should not be taken into account …”.

Checking of English style with a native speaker is recommended

Author Response

Dear Reviewer,

We took the time to create a separate word file to answer your concerns. Please, see the attachment. 

Thank you again for your time and understanding, we appreciate it very much.

Kind regards,

The authors

Reviewer 2 Report

The proposed paper seems interesting. However, I have some doubts that the authors should clarify/discuss: 

- the title is a bit misleading. The algorithm used can be applied, in general, to other fields. The authors should modify the title removing the "histological samples" portion. They used histological samples just to test and validate their work; they could have used any other "2D" samples. The authors should have a more generalized discussion about the applicability of the method and mention that they will use the samples to test/validate its performances.

- some minor grammar typos; see highlighted. 

-the calibration work for static imaging (no samples, illuminated and dark background). However, the authors should explain if that calibration works also when the microscope stage moves; the proposed calibration may not sufficient to remove artifacts from the images taken during "dynamic conditions"; discussion and/or experiment should be shown. 

- some images have their label (a, b, etc.) partially covered and should be fixed.  

Author Response

(The authors gave the same response as above.)

Round 2

Reviewer 2 Report

Thank you for addressing the comments and improving the paper.